# Research data management needs assessment for social sciences graduate students: A mixed methods study

**Xuan Zhou**[1], **Zhihong Xu**[2]*, **Ashlynn Kogut**[3]

**1** Department of Research Data Services, University Libraries, Texas State University, San Marcos, Texas, United States of America, **2** Department of Agricultural Leadership, Education, and Communications, Texas A&M University, College Station, Texas, United States of America, **3** Department of Teaching, Learning, and Culture, Texas A&M University, College Station, Texas, United States of America

* xuzhihong@tamu.edu

**Data Availability Statement:** All data files is available from the Texas Data Repository. The data is available at https://doi.org/10.18738/T8/YQ1XDX.

## Abstract

The complexity and privacy issues inherent in social science research data makes research data management (RDM) an essential skill for future researchers. Data management training has not fully addressed the needs of graduate students in the social sciences. To address this gap, this study used a mixed methods design to investigate the RDM awareness, preparation, confidence, and challenges of social science graduate students. A survey measuring RDM preparedness and training needs was completed by 98 graduate students in a school of education at a research university in the southern United States. Then, interviews exploring data awareness, knowledge of RDM, and challenges related to RDM were conducted with 10 randomly selected graduate students. All participants had low confidence in using RDM, but United States citizens had higher confidence than international graduate students. Most participants were not aware of on-campus RDM services, and were not familiar with data repositories or data sharing. Training needs identified for social science graduate students included support with data documentation and organization when collaborating, using naming procedures to track versions, data analysis using open access software, and data preservation and security. These findings are significant in highlighting the topics to cover in RDM training for social science graduate students. Additionally, RDM confidence and preparation differ between populations so being aware of the backgrounds of students taking the training will be essential for designing student-centered instruction.

## Introduction

Technology has transformed the collection, analysis, storage, and sharing of research data, making knowledge of research data management (RDM) essential for researchers. RDM consists of the activities done by a researcher to ensure that the research data collected is findable, protected, preserved, and shareable [1]. RDM knowledge and skills are essential for researchers to work successfully. When informational data is incorporated into research publications, effective RDM practices enhance research outcomes [2].

**Funding:** This work was supported by T3: Texas A&M Triads for Transformation [grant numbers 1762, 2020], Texas A&M University Institutional Funding. The funders had no role in study design, data collection and analysis, decision to publish, or preparation of the manuscript. There was no additional external funding received for this study.

**Competing interests:** The authors have declared that no competing interests exist.

Graduate students are learning the research process and norms of their discipline, which makes learning RDM at the same time ideal for preparing graduate students for the current research environment. To meet this need, higher education institutions and academic libraries have developed training for graduate students [3–6]. Due to the unique characteristics of social science research data, RDM training should be designed with a thorough understanding of the needs of social science graduate students. However, the available literature provides limited information about the RDM needs of social science graduate students.

## Importance of RDM for social science graduate students

Studies have demonstrated the value of developing the RDM skills of graduate students as they typically have active involvement with research data with some guidance from their academic advisors [7–9]. Most of the time, graduate students are expected to independently handle all data management tasks for their own studies [10]. Therefore, understanding graduate students' needs for developing their RDM knowledge and practices is crucial, since they will be the future generation of researchers [11].

The needs and practices of RDM vary among research disciplinary domains due to the distinct nature of research data collected [12, 13]. Graduate students in the social sciences have particular research needs. Data generated from social science studies might include observations of individual behaviors, assessments of an individual's psychological situation, and surveys of participants' perceptions. This data may consist of images, audio, video recordings, or spreadsheets with a mix of quantitative and qualitative data, in addition to secondary data collected by others [11, 14]. Therefore, graduate students in the social science disciplines need to develop their understanding of RDM across the data life cycle as well as expand their RDM skills to work with the complex, nuanced data types and formats in their research field.

## Importance of needs assessment for social science graduate students

Over the past ten years, researchers have encountered many challenges related to managing data [15–19]. Academic librarians have noticed these issues, and in response, many academic libraries have established RDM services and conducted research on RDM needs and best practices [20–23].

As indicated by Goben and Griffin [20], conducting a needs assessment is an efficient approach to identify the areas where researchers face obstacles in RDM and, consequently, the areas best suited to address with academic library services. To better understand the needs of researchers, some studies investigated researchers' current RDM practices and needs [9, 24–28]. Although these studies have sought to identify the RDM skills and needs of researchers and graduate students across disciplines, these needs assessments are not sufficient to inform academic libraries in planning RDM services specifically for social science graduate student researchers who are in the early phases of doing research.

RDM instruction for social science graduate students is still in development, as only a few studies reported RDM instruction specifically designed for this population [22, 23]. Due to the nature of the data generated from research projects, graduate students in the social sciences may have distinct needs for RDM instruction or challenges in RDM practices compared with students in other disciplines (e.g., STEM disciplines) [14]. There is little information in the literature about how academic libraries can effectively prepare social science graduate students for RDM. Research, particularly in the social science fields, does not provide sufficient evidence regarding the RDM needs and challenges from students' viewpoints [20]. In order to ensure that social science graduate students can receive effective disciplinary tailored RDM

services from academic libraries, understanding students' needs, preparedness, and challenges related to RDM is essential.

## Research questions

The present study aims to fill this research gap by investigating social science graduate students' perceptions of their RDM needs, confidence, preparedness, and challenges to identify areas to target when creating RDM training or professional development opportunities. We attempted to address the following research questions:

1. What experiences do social science graduate students have with RDM, such as knowledge of available resources, creation of data management plans, data sharing practices, and preparation for RDM?

2. Does the confidence and preparation of social science graduate students for RDM vary by their race/ethnicity, years in the program, and citizenship?

3. What challenges do social science graduate students face with RDM?

## Methods

### Research design

As a public, non-profit, R1, higher education institution in the United States, Texas A&M University Libraries serves approximately 14,000 graduate/professional scholars in over 270 graduate-level programs. The University Libraries started to provide a series of RDM services in 2020 and planned to tailor its RDM-related instruction and services to address general research needs as well as the needs of individual disciplines. Existing studies reported that providing disciplinary-based RDM training would maximize the audience's RDM knowledge and skills [6, 29, 30]. Graduate researchers are at the early stages of their research careers and usually have less research experience than university faculty and professional researchers. Therefore, identifying graduate researchers' RDM preparedness and needs is key to developing a successful disciplinary-based RDM training program.

A mixed method design was used for this study. First, a survey through Qualtrics was disseminated to the graduate students in the School of Education at Texas A&M University. The survey aimed to assess social science graduate students' experience with RDM, preparedness for RDM, and their needs for further training related to RDM. The survey was designed based on the previous literature [9, 25, 26]. Prior to distribution, the survey was evaluated by a group of experts, which included two librarians from the RDM service department and one library and information science scholar. The survey draft was piloted with three graduate students from the target population for further validation. The final version survey was created in Qualtrics and distributed by bulk email to graduate students in the School of Education. Follow-up in-depth interviews were conducted with ten graduate students to collect more information regarding their data awareness, knowledge of RDM, and challenges related to RDM practices across the research data life cycle.

### Participants

After receiving Texas A&M University institutional review board approval (IRB2020-1180D), the authors used the institution's bulk email to recruit participants. Before the survey, participants were required to verify that they were at least 18 years old and were informed about what experiment they would be participating in. Then, all the participants signed a consent

**Table 1. Demographic information of participants (N = 98).**

| Characteristics | | N | % |
|---|---|---|---|
| Years in program | 2 years or less | 64 | 65.31 |
| | > 2 years | 34 | 34.69 |
| Degree pursuing | Doctoral | 81 | 82.65 |
| | Master | 17 | 17.35 |
| Gender | Female | 81 | 87.10 |
| | Male | 12 | 12.90 |
| Race/ethnicity | White & Asian | 76 | 77.55 |
| | Under-represented minorities | 22 | 22.45 |
| Citizenship | U.S citizen or Permanent Resident | 61 | 62.24 |
| | International student | 37 | 37.76 |

form through Qualtrics before their participation. One hundred and fifty-five responses were received. After removing responses that lacked respondent's demographic information and duplicates, 98 valid responses were collected from the survey distribution (Table 1). A total of 10 graduate students from the School of Education were randomly selected and recruited for an interview. Their department affiliations, years of research experience, and types of research data they are currently working with were diversified. The respondents' demographic information is displayed in Table 1.

## Measures

After the review and pilot test of the survey questions, a total of 13 items were distributed to measure graduate students' RDM preparedness and training needs. The survey consisted of five topics: knowledge of available resources, experience with data management plans (DMPs), experience with data sharing, preparedness in RDM, and the needs for RDM professional development.

The section on graduate students' knowledge of available resources contained three items. This section was designed to understand how knowledgeable the graduate students were about the RDM services and resources at the university. The experience in the data management planning section contained two items. In this section, students were assessed on whether they had experience writing a DMP. The experience with data sharing section covered four items, including assessing participants' familiarity and use of data repositories. Participants were further assessed on their preparedness and training needs related to RDM by five items. Two items asked about participants' training needs in RDM and their confidence level in dealing with RDM on a rating scale of zero to 100 points. The other three items assessed participants' preparedness for RDM on a 5-point Likert scale from very unprepared to very prepared. The full survey is included in S1 Appendix.

The interviews were guided by an interview protocol consisting of semi-structured and probing questions. The questions covered multiple aspects of RDM, such as data awareness, DMPs, current practice of data organization, preservation, data sharing for their research project, and challenges in their daily RDM practice. Participants were randomly selected from a pool of respondents who participated in a four-hour RDM instruction session focused on the essential aspects of RDM practice.

## Data analysis

STATA 17 was used to analyze the survey data. Descriptive statistics were analyzed to evaluate the current status of graduate students' knowledge of available resources, experience with

DMPs, experience in data sharing, and preparedness and needs for professional development in RDM. We also hypothesized that there might be statistically significant differences in students' overall confidence with RDM and preparedness for RDM by race/ethnicity, years in their graduate program, and citizenship. One-way ANOVAs were tested for both hypotheses. To test these hypotheses, we used the composite score of the four confidence level survey questions and the three survey questions addressing students' RDM preparedness level. Students' years in their graduate program were coded into two groups with *1 = more than 2 years* and *0 = 2 or fewer years*. Similarly, students' race/ethnicity was coded as *1 = White & Asian*, and *0 = Under-represented Minorities* including African American, Hispanic or Latino, Native American, and Pacific Islanders. Participants' citizenship was coded as *1 = U.S citizen or Permanent Resident* and *0 = International student*. Before conducting the ANOVA, assumption checking was performed and met.

Conventional content analysis was used to analyze the interview data to identify students' experiences with DMPs and data sharing, their needs for professional development, their needs for RDM, and their confidence level and preparedness for RDM [31].

## Results and discussion

### Graduate students' experiences with RDM

**Knowledge of available RDM resources.** The survey assessed participants' knowledge of available RDM resources for the university community (Table 2). Regarding graduate students' awareness of the RDM Service (RDMS) at the University Libraries, more than half (53%) had never heard about the service. About 34% of the respondents said they had heard of the RDMS, and 13% of the respondents shared that they had probably heard of this service before but were unsure.

For how the respondents heard of the RDMS, the majority of the respondents selected "n/a" (68.37%), meaning that there might be limited marketing and advertising making them aware of the available resources. On the other hand, 5.10% selected workshops, and 20.41% said they heard from other faculty, students, and/or staff. A few also heard about this service from the library's website (2.04%). Additionally, 4.08% of the respondents selected more than one way that they had heard about this service (Fig 1).

The survey also asked the respondents whether they had experience working with librarians on RDM. More than 60% of participants responded that they did not have any experience, and 22.45% of participants shared that maybe they will work with a librarian on RDM in the future. 14.29% of the respondents said they had experience working with a librarian on RDM, whereas 3.06% were unsure whether they had this experience. The follow-up interviews provided additional context to the survey findings. Only two of the interviewees reported working with a

**Table 2. Knowledge of available resources.**

| Items | Category | N | % |
|---|---|---|---|
| Heard of RDMs | Yes | 33 | 33.67 |
| | No | 52 | 53.06 |
| | Probably | 13 | 13.27 |
| Experience working with a librarian on RDM | Yes | 14 | 14.29 |
| | No | 59 | 60.20 |
| | Probably | 3 | 3.06 |
| | Maybe in the future | 22 | 22.45 |

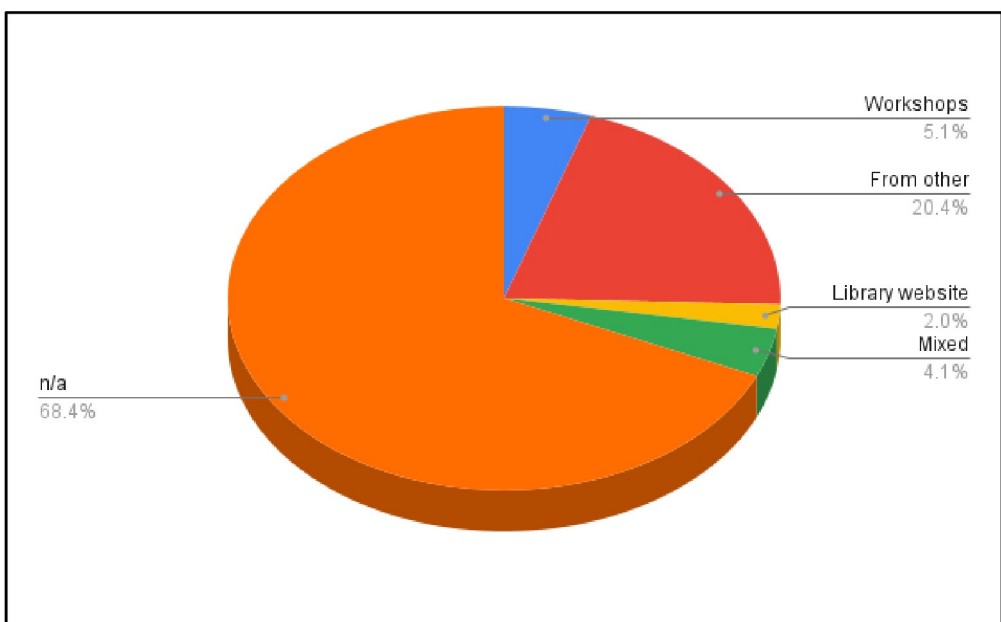

**Fig 1. Distribution of the method of hearing about RDMS.**

librarian for secondary data collection, when conducting a systematic review and for data publication issues. The majority of the interviewees had no prior experience collaborating on RDM with an academic librarian. As a result, social science graduate student interviewees were unfamiliar with the institutional RDM services, which is consistent with the survey results. Therefore, there is a need to increase social science graduate researcher awareness of available RDM resources and encourage the use of the resources.

Beginning in 2020, the University Libraries established a new unit to meet the RDM needs of the research community. However, more than half of the respondents were unaware of this service. These findings were consistent with previous literature that early career researchers did not have adequate awareness of existing library and institutional RDM services [32]. The lack of awareness shown by social science graduate students is a critical issue for the library's RDM unit. RDMS is an integrated service requiring different units to work together. RDM consists of activities across the data lifecycle, including data collection, storage, privacy, sharing, preserving, and reusability [33]. Additionally, RDM requires awareness of technical, ethical, and legal issues as well as government policies regarding the sharing of research data [34]. A successful RDMS needs all the relevant organizations—libraries, research supporting services, and information technology—in a university to collaborate seamlessly to raise the research community's awareness of RDMS. In this collaboration, the academic departments' involvement is especially important, as 20.41% of the graduate students heard about RDMS from other faculty, students, or staff. Academic units have direct influence on students and are one avenue for marketing. For example, representatives of the RDMS can speak or present during graduate students' courses to advocate for the importance of RDM, and therefore, expose the RDMS to this important group.

**Experience with DMPs and data sharing.** The survey evaluated participants' experience with DMPs by asking whether they had ever written a DMP for a grant (Fig 2). A DMP describes data that will be acquired or produced during research. It will include information on how the data will be produced, managed, stored, and shared; what metadata standards

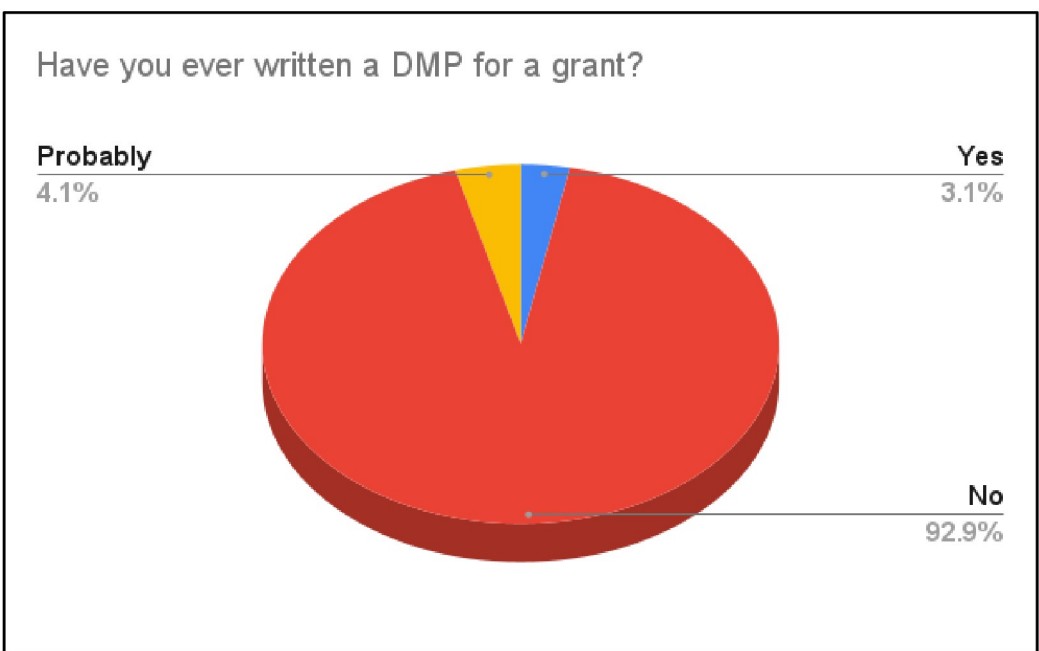

**Fig 2. Experience with DMPs.**

researchers use; and how data will be handled and protected during and after the completion of the project. 92.86% of respondents had never written a DMP before, whereas 3.06% had this experience. Additionally, 4.08% of the respondents shared that they probably have written a DMP for a grant.

In the interviews, students reported a lack of awareness of standard practices for DMP. As noted by most interviewees, there might be some procedures or processes regarding data management in their research team; however, they were not aware of whether those are consistently standardized as plans for data management. For example, one interviewee shared, "*I don't think we have some systematic procedure or process for DMP. But sometimes we are trying to make it standardized.*" Another student said, "*Maybe I am not aware of it. From my experience, there is a set of practices, but I think there are more like sorts of best practices to organize the data.*" These findings provide additional support for the conclusions regarding the necessity and importance of encouraging the development and adoption of a standardized procedure in DMP for graduate researchers.

Experience with grant writing is of critical importance for graduate students [35], and a DMP is an important part of grant writing. Involvement with grant writing will make the next-generation researcher more prepared for a future career as a faculty member. A good DMP can provide evidence for research in conjunction with published results. It can increase research's impact and visibility through data citation [36, 37]. The survey and interviews showed that more than 90% of the respondents never had experience with writing a DMP, which should be noted by the graduate programs in higher education. When designing a graduate degree program, RDM courses should be integrated into the core curriculum since it is an indispensable and crucial skill for the next-generation of researchers.

In terms of participants' experience in data sharing, their familiarity with research data repositories was first assessed. More than half of the respondents (59.18%) reported not being familiar with any research data repository, followed by 28.57% who reported familiarity with a

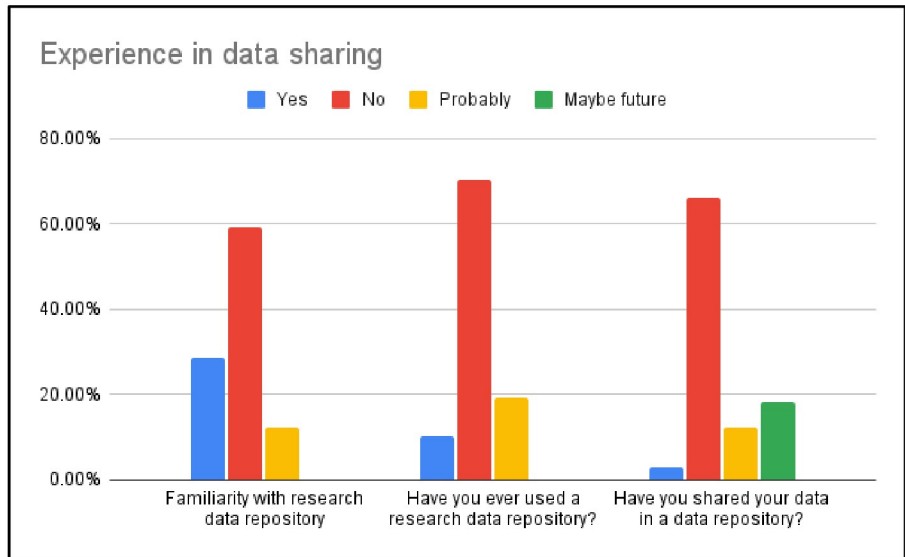

**Fig 3. Experience in data sharing.**

data repository, and 12.24% who were unsure what a research data repository was. We also assessed whether the students ever used a research data repository. The majority of the respondents did not have experience using a data repository (70.41%), 10.20% said they had used a research data repository previously, and 19.39% were not sure whether they had this experience or not. Regarding graduate students' experience sharing data via a data repository, 66.33% of them responded that they never shared data in a data repository, and 18.37% said they might share data via a data repository in the future. Only three of the respondents (3.06%) had experience sharing data via a data repository. Additionally, 12.24% of the respondents selected probably, indicating that they were not sure whether they had this experience (Fig 3).

The follow-up interviews confirmed the findings from the survey that the majority of interviewees did not have experience sharing data. Interviewees pointed out that there was currently no obligation for data sharing in their fields. Additionally, none of the interviewees demonstrated familiarity with data repositories. One student did mention that a particular grant-funded study required her team to disclose the data after the study was finished. However, she was not aware of any data repository and neither were her peer graduate students on that research team. She stated, *"We do not have a specific role for data sharing, but I did learn some data repositories after attending your RDM workshop."* This implies that graduate students need more training about data repositories and the best practices for preserving and sharing data.

Robustness, reproducibility, reliability, and transparency are increasingly recognized as foundational for impactful research [38, 39]. To realize this, data sharing is necessary and significant for quality research. By creating journal level data sharing policies, some journals and publishers have tried to increase the prevalence of data sharing [40]. However, students' experience with data sharing is very limited. This again highlights the need for graduate programs to increase emphasis on RDM education, which could involve collaboration between academic libraries and other important stakeholders (e.g., departments, the whole research community, *etc.*).

**Needs of professional development and confidence in dealing with RDM.**   Our survey also assessed social science graduate students' training needs and their confidence level in

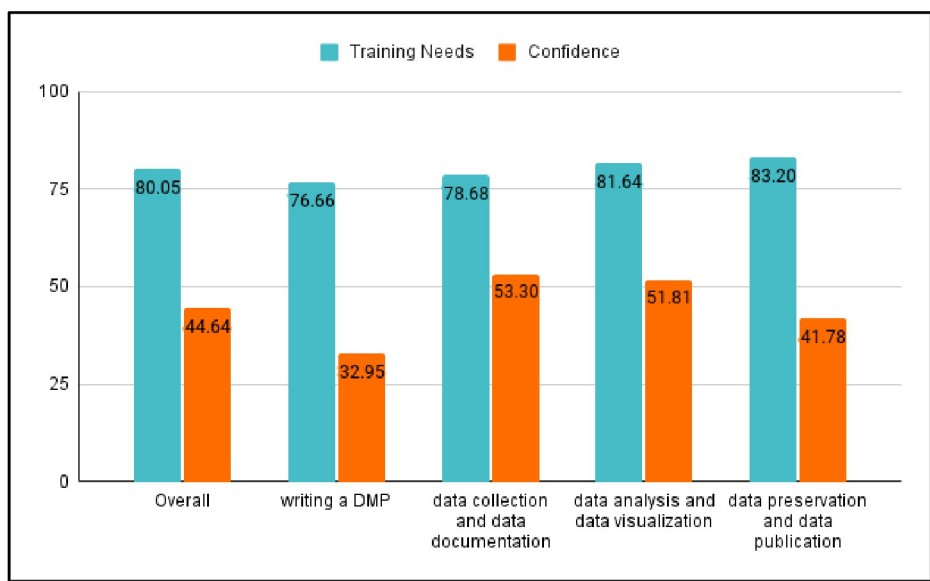

**Fig 4. Mean score comparison between graduate students' training needs and confidence level in dealing with RDM.**

performing RDM. Fig 4 shows that students had a low degree of confidence in their capacity to deal with RDM and needed significant growth in RDM-related knowledge and skills. Graduate students' reported their overall needs for RDM training as 80.05 (SD = 18.71) out of 100, while their confidence level in handling RDM was 44.64 (SD = 25.49) out of 100. More specifically, the mean score for each sub-item of student needs in RDM training ranged from 78.66 to 83.20. Additionally, the mean score for each component of the students' confidence level ranged from 32.95 to 53.30. From our survey data, we noticed that students' needs for professional development in RDM corresponds to their confidence level in RDM. Because they are not confident about their RDM knowledge, they have a high need for RDM instruction and training. This indicates that the social science graduate students are aware of the importance of RDM when conducting research.

Results from the interviews further supported the need for graduate students to develop skills in RDM. Several themes from the interviews corresponded to the survey data, including the need for instruction in data documentation and data organization when working on teams, data analysis with open access software, and data preservation.

Regarding data organization, students demonstrated basic knowledge of the practice of data version tracking, such as naming a file by its content and using the year with a date or numbers to track the versions. For example, one student shared that *"each file and video was named after by a sequence of numbers, and alphabet letters and also by class of the unit."* Another student shared that *"I use a file naming system to assign names to my data files, save each version that I do differently, like V1, V2, or V3 at the end of the name."* Regarding the data record and familiarity with metadata, almost all students reported not having much experience with metadata or not being familiar with the terms. Only one student shared that she uses a glossary of variables shared by team members while recording data. Most of the time, students do not record their data progressively in a timely manner.

Overall, students demonstrated a lack of consistent data documentation and organization, especially when collaboratively working in a research group. As a student reported, *"usually*

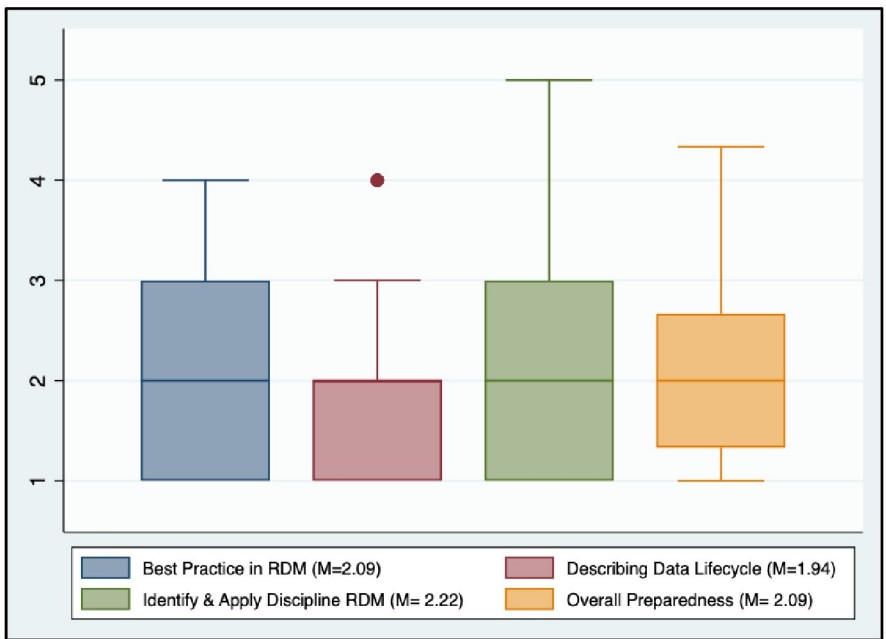

**Fig 5. Preparedness for RDM.**

*what I'll end up doing is looking at one file with a date, and then I can see which is the most recent. I know it is inefficient, but I guess that works for me personally, not sure for other team members."* Another student also indicated that even though she always put a year and either a month, day, or semester on the file she worked with, she couldn't figure out the identifiers created by others for data version tracking.

Additionally, only a few students were knowledgeable about data quality control. Students in the social sciences are frequently involved with human subjects and collect data on an individual basis. The variability of individual biometrics (e.g., facial structure, fingerprints, signatures) can impact data quality as well as data security [41]. However, the majority of students reported not being aware of this issue. Therefore, there is a need to develop social science graduate students' independent data documentation and organization skills. Additionally, graduate students should learn how to integrate these skills when working collaboratively.

Concerning students' perspectives about data analysis and visualization with open access tools (e.g., R programming), most students rely on proprietary software (e.g., SPSS and STATA), since they are instructed to use proprietary software in department-offered statistics courses. Students are interested in using open access software for future research but lack appropriate resources in learning how to utilize them.

Although most students reported storing their data in either Google Drive or Microsoft OneDrive and sharing data with other team members, few of them considered preserving their data for long-term use. Moreover, only a few participants have considered data security issues, as human research data can contain sensitive information. Therefore, it is necessary to improve graduate students' awareness of security measures and long-term data preservation for reuse and sharing.

**Students' preparedness for RDM practices.** The survey also assessed graduate students' preparedness for RDM practices (Fig 5). Overall, graduate students' preparedness for RDM was 6.26 (SD = 2.69) out of 15. As seen at the item level measured by the 5-point-Likert scale, students were prepared to describe best practices in RDM at an average level of 2.09

**Table 3. RDM confidence by race/ethnicity, years in the program, and citizenship.**

| Confidence in RDM | SS | df | F | $\eta^2$ |
|---|---|---|---|---|
| Race/Ethnicity | 993.56 | 1 | 1.80 | .018 |
| Year in program | 592.85 | 1 | 1.07 | .011 |
| Citizenship | 3697.63 | 1 | 7.08** | .07 |

Note.
*p < .05,
**p < .01,
***p < .001

(SD = 1.05), describe the research data lifecycle at 1.94 (SD = 1.01), and find and apply discipline-appropriate RDM approaches/principles to their research project 2.22 (SD = 1.05). This shows that students' readiness for RDM practices is still lacking, and students require further training in RDM.

## Difference in RDM confidence by participants' race/ethnicity, years in the program, and citizenship

**Difference in RDM confidence level.** One-way ANOVAs were used to determine whether respondents' overall confidence level about RDM varies by their race/ethnicity, research experience, and citizenship (Table 3 and Fig 6). Results showed that respondents who are U.S. citizens or permanent residents (domestic students) had statistically significant higher confidence levels in doing RDM than international students (F = 7.08, p < .001) with a small effect size ($\eta^2 = .07$). However, no differences were found between White/Asian graduate students and underrepresented minorities (F = 1.80, p = 0.1824). A similar result was found

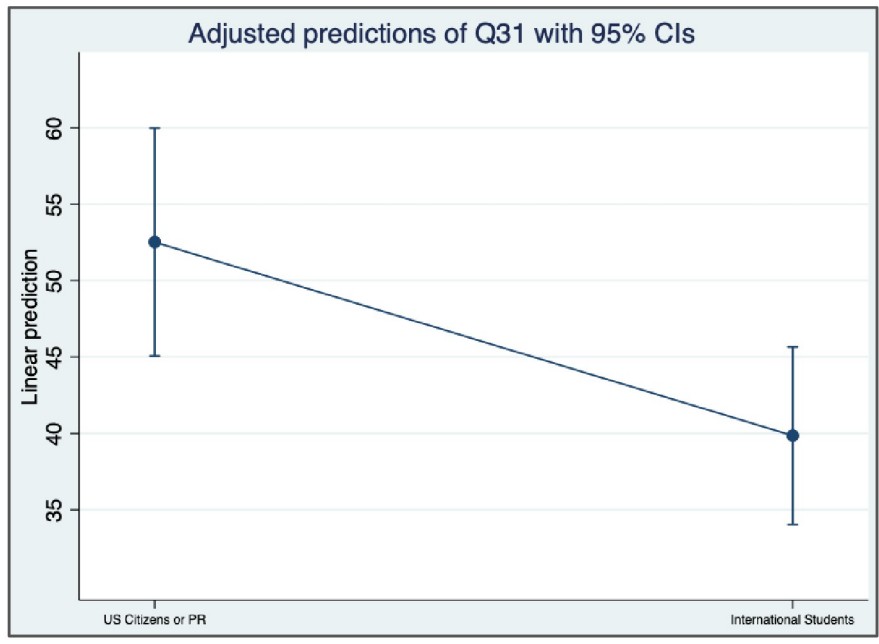

**Fig 6. Confidence level difference between U.S. citizens and international students.**

between students who had less than 2 years of studying in the current graduate program and students who had more than two years of studying, indicating no significant differences (F = 1.07, p = 0.3039).

The difference in confidence found between domestic students and international students aligns with prior research examining the confidence levels of students. Zhao and Mawhinney [42] found that native English-speaking students had more confidence in using library resources and services, even without attending a training session, than native Chinese-speaking students. International graduate students are also navigating the language of their host country and discipline as well as learning a new culture both in the United States and in academia [43, 44]. Navigating these circumstances could also contribute to student's confidence levels regarding RDM. It is also important to consider that differences could exist between groups of international students. For example, one study found that female international business students were more confident in their information literacy skills than male students [45]. Future research can continue to explore whether differences in RDM confidence exist between different groups of international students. One way to help international graduate students build confidence in utilizing RDM is to use inclusive instruction that integrates culturally diverse perspectives [46]. Because overall both domestic students and international students lacked confidence in RDM, an inclusive approach to RDM instruction coupled with the integration of diverse perspectives about research data could help increase confidence of all graduate students when learning RDM.

**Difference in RDM preparedness.** One-way ANOVAs were also conducted to evaluate whether graduate students' overall RDM preparedness varies by race, research experience, and citizenship (Table 4, Figs 7, and 8). Results demonstrated that White/Asian graduate students had a statistically significant higher level of preparedness in RDM than underrepresented minorities (F = 6.54, p < .05), with a small effect size ($\eta^2$ = .06). Further, graduate students who were U.S citizens or permanent residents showed statistically significant higher levels of RDM preparedness than international students (F = 3.73, p < .05) with a small effect size ($\eta^2$ = .04). However, no group differences were found by participants' years in the current program (F = 2.62, p = 0.1089).

Most RDM training offered by academic libraries has focused on faculty, researchers, and librarians [23], so graduate students' limited RDM preparedness is not surprising. The findings that underrepresented minorities and international students had less preparation for implementing RDM than white/Asian graduate students are important considerations for marketing RDM support as well as designing inclusive RDM training. Training and instruction from libraries targeted to international graduate students have largely focused on general information literacy skills [47]. In fact, one recent study examining information literacy instruction for international graduate students, excluded the topic of RDM from the session to focus on

**Table 4. RDM preparedness by participants' personal characteristics.**

| Preparedness | SS | df | F | $\eta^2$ |
|---|---|---|---|---|
| Race/ethnicity | 4.96 | 1 | 6.54* | .06 |
| Year in program | 2.07 | 1 | 2.62 | .03 |
| Citizenship | 2.91 | 1 | 3.73* | .04 |

Note.

*p < .05,

**p < .01,

***p < .001

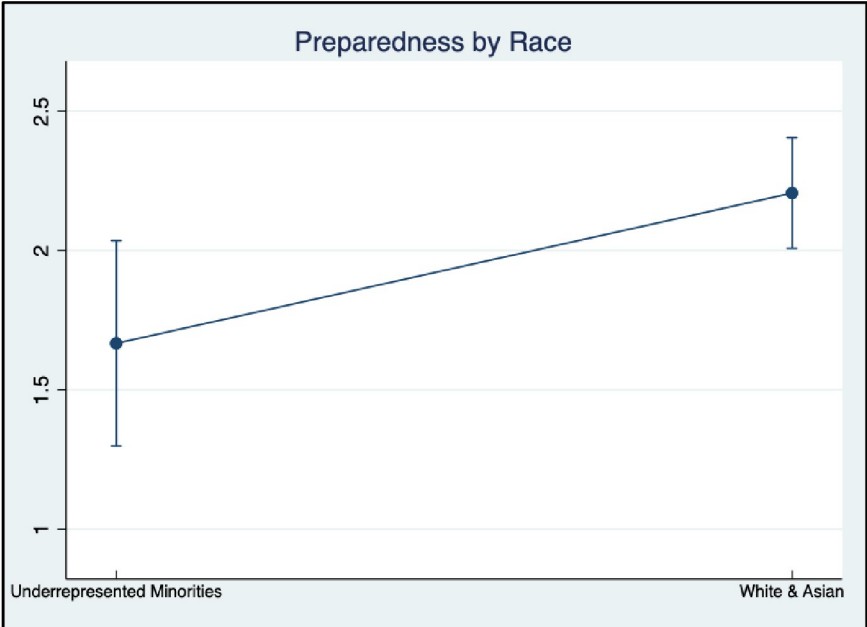

**Fig 7. RDM preparedness by race/ethnicity.**

searching and locating scholarly information [48]. International graduate students are unfamiliar with the academic library [46] and/or lack knowledge of what the library offers [42], so to further develop international graduate students as researchers it is important to make this population aware of the advanced research support offered by libraries and other on-campus services.

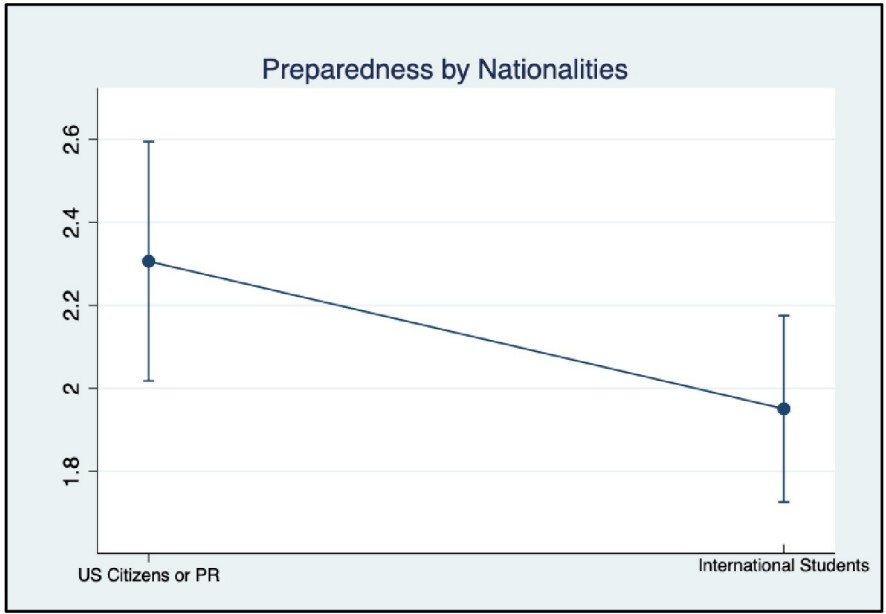

**Fig 8. RDM preparedness by nationality.**

## Challenges

From the interview data, we identified several challenges that social science graduate students experience with RDM. First, interviewees noted graduate students face challenges in collecting and gathering data. Students who primarily worked on original data collection lacked skills in participant recruitment, resulting in a lack of respondents or data attrition in the post-measure stage of a study. Moreover, students indicated that if collaborating with another research team, different data collecting conventions between institutions may result in difficulties in understanding the data collected by the other team. On the other hand, students who use secondary data for their own research discussed that they lacked knowledge of and access to resources for open-access datasets that suited their research topic.

In terms of the difficulties that the graduate students faced in the data analysis and modeling aspects, they did not feel competent in determining appropriate statistical models for their data and demonstrated a lack of ability to interpret the results of data analysis. One student shared:

> *"Interpreting the results and modeling are challenging, especially when you are dealing with interdisciplinary research projects. I think for us it's typically just making sense of it. We can look for patterns, and we can visualize them however we want. But unless we're able to explain it accurately."*

Students also reported that, since social science research usually involves interdisciplinary projects with both qualitative and quantitative data and is not very objective, cleaning the data and coding for variables were challenging for students with limited research experience.

The interviews also revealed challenges in students' experience with data storage. Even though most interviewees shared that they stored their data in Google Drive or Microsoft OneDrive, it is still hard for them to locate the correct file. Meanwhile, the security of data storage was another concern among graduate students. However, they were unaware of effective ways to backup their data.

## Conclusion

RDM training for graduate students in the social sciences is an under-researched area. This study assessed social science graduate students' perceptions of their RDM needs, confidence, preparedness, and challenges. While this study's findings provide a starting point for developing targeted training for social science graduate students, tailored RDM training should be based on the students' knowledge of data core competencies. Data core competencies are the skills needed to access, evaluate, manipulate, preserve, and use data. Therefore, future research should aim to design standardized data core competency measures to help researchers and practitioners evaluate students' data core competency levels. Future research should also examine the RDM needs of graduate students in social science disciplines outside of education (e.g, psychology, sociology, economics) to determine if other social sciences disciplines have different needs related to RDM.

Our findings highlight important practical aspects to consider when developing awareness of on-campus RDM services, determining the content to cover during training sessions, and designing inclusive learning experiences that acknowledge the unique circumstances of students. Social science graduate students were unaware of RDM services available on campus. Those who were aware of RDMS often heard about it through word of mouth. For students to make use of RDMS, additional advertising and marketing needs to be developed specifically for this group. To better prepare this population for effective RDM practices, RDM should be

integrated throughout the graduate program. Higher education administrators should consider requiring RDM training for graduate students and provide support for this initiative.

Overall, students were not prepared for RDM, indicating that additional training is needed. Topics to cover during instruction sessions include data management plans, using a data repository, sharing research data, data documentation, data organization, and data storage for long-term use. The implications of proprietary software on RDM and of storing human subjects data in cloud-based services like Google Drive or Microsoft OneDrive are additional topics to include that specifically address social science data. How to integrate RDM practices when working in research teams is another topic to discuss with graduate students.

The participants also identified challenges regarding research and data collection in the social sciences utilizing human subjects data and secondary datasets. These challenges include collecting and gathering data, data analysis and modeling, mixing quantitative and qualitative data, cleaning the data, coding for variables, locating correct files, and backing up/preserving data. While some of the challenges identified are better addressed in research methods courses than stand-alone RDM training, these challenges can be used to provide context for developing RDM training.

Finally, when covering the RDM content areas identified, it is important to design learning environments that address the needs of international students and underrepresented minority students. These student populations were found to have lower confidence levels and less preparation for RDM. Utilizing these findings when developing and delivering RDM training sessions will help social science graduate students develop the skills needed to manage research data for projects in graduate school and in their future careers.

## Supporting information

**S1 Appendix. Survey instrument.**
(DOCX)

## Author Contributions

**Conceptualization:** Xuan Zhou, Zhihong Xu.

**Data curation:** Xuan Zhou.

**Formal analysis:** Xuan Zhou.

**Funding acquisition:** Zhihong Xu, Ashlynn Kogut.

**Investigation:** Xuan Zhou, Zhihong Xu.

**Methodology:** Xuan Zhou, Zhihong Xu, Ashlynn Kogut.

**Project administration:** Xuan Zhou, Zhihong Xu.

**Resources:** Zhihong Xu.

**Software:** Xuan Zhou.

**Supervision:** Zhihong Xu.

**Validation:** Xuan Zhou, Zhihong Xu, Ashlynn Kogut.

**Visualization:** Xuan Zhou.

**Writing – original draft:** Xuan Zhou, Zhihong Xu, Ashlynn Kogut.

**Writing – review & editing:** Xuan Zhou, Zhihong Xu, Ashlynn Kogut.

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
