## [Decision Letter · Decision Letter 0]

26 Dec 2022

PONE-D-22-30504Research data management needs assessment for social sciences graduate students: A mixed methods studyPLOS ONE

Dear Dr. Xu,

Thank you for submitting your manuscript to PLOS ONE. After careful consideration, we feel that it has merit but does not fully meet PLOS ONE’s publication criteria as it currently stands. Therefore, we invite you to submit a revised version of the manuscript that addresses the points raised during the review process. Please submit your revised manuscript by Feb 09 2023 11:59PM. If you will need more time than this to complete your revisions, please reply to this message or contact the journal office at plosone@plos.org. Please include the following items when submitting your revised manuscript:A rebuttal letter that responds to each point raised by the academic editor and reviewer(s). You should upload this letter as a separate file labeled 'Response to Reviewers'.A marked-up copy of your manuscript that highlights changes made to the original version. You should upload this as a separate file labeled 'Revised Manuscript with Track Changes'.An unmarked version of your revised paper without tracked changes. You should upload this as a separate file labeled 'Manuscript'.If applicable, we recommend that you deposit your laboratory protocols in protocols.io to enhance the reproducibility of your results. Protocols.io assigns your protocol its own identifier (DOI) so that it can be cited independently in the future. For instructions see: https://journals.plos.org/plosone/s/submission-guidelines#loc-laboratory-protocols. Additionally, PLOS ONE offers an option for publishing peer-reviewed Lab Protocol articles, which describe protocols hosted on protocols.io. Read more information on sharing protocols at https://plos.org/protocols?utm_medium=editorial-email&utm_source=authorletters&utm_campaign=protocols.

We look forward to receiving your revised manuscript.

Kind regards,

Omar Mohammad Ali Khraisat, Associate Professor

Academic Editor

PLOS ONE

Journal Requirements:

3. Peer review at PLOS ONE is not double-blinded (https://journals.plos.org/plosone/s/editorial-and-peer-review-process). For this reason, authors should include in the revised manuscript all the information removed for blind review.

"This work was supported by T3: Texas A&M Triads for Transformation [grant numbers 1762, 2020], Texas A&M University Institutional Funding."

"This work was supported by T3: Texas A&M Triads for Transformation [grant numbers 1762, 2020], Texas A&M University Institutional Funding."

7. We note that you have stated that you will provide repository information for your data at acceptance. Should your manuscript be accepted for publication, we will hold it until you provide the relevant accession numbers or DOIs necessary to access your data. If you wish to make changes to your Data Availability statement, please describe these changes in your cover letter and we will update your Data Availability statement to reflect the information you provide.

Reviewers' comments:

Reviewer's Responses to Questions

**Comments to the Author**

1. Is the manuscript technically sound, and do the data support the conclusions?

Reviewer #1: Yes

Reviewer #2: Yes

2. Has the statistical analysis been performed appropriately and rigorously? 

Reviewer #1: I Don't Know

Reviewer #2: Yes

3. Have the authors made all data underlying the findings in their manuscript fully available?

Reviewer #1: No

Reviewer #2: Yes

4. Is the manuscript presented in an intelligible fashion and written in standard English?

Reviewer #1: Yes

Reviewer #2: Yes

5. Review Comments to the Author

Reviewer #1: It is a well-structured work that helps to understand some of the challenges faced by new researchers who, sometimes, put their investigations at risk due to a lack of these skills.

About data - The data should be provided as part of the manuscript or its supporting information, or deposited to a public repository. If there are restrictions on publicly sharing data, those must be specified.

Reviewer #2: The manuscript addresses a significant problem in the selected area of research. Furthermore, the contents are managed coherently. However, the following observations need redressal:

1. Instead of mentioning 'Research Question 1" in the subheading in Results and Discussion, thematic headings would make them more comprehensible. This applies to all the subheadings in this section.

2. Discussion on the interview data is limited and needs further instances, elaboration, and due support from the reviewed literature.

3. It seems that the data has not properly been triangulated.

4. Recommendations for future studies are not specified.

5. The authors might be advised to get the manuscript proofread by an expert in the English language.

6. PLOS authors have the option to publish the peer review history of their article (what does this mean?). If published, this will include your full peer review and any attached files.

Reviewer #1: No

Reviewer #2: **Yes: **Rooh Ul Amin

Professor

Department of English

Associate Dean, Faculty of Arts and Social Sciences

Foundation University

Islamabad

---

## [Author Response · Author response to Decision Letter 0]

30 Jan 2023

Comment Response

Academic editor’s

We formatted our manuscript following the PLOS ONE style templates. 

We indicated that “all the participants signed the written consent form through Qualtrics before their participation” in the Participants sub-section of the Methods section, the disclosure statement and in the submission system. 

3. Peer review at PLOS ONE is not double-blinded (https://journals.plos.org/plosone/s/editorial-and-peer-review-process). For this reason, authors should include in the revised manuscript all the information removed for blind review.

Identifying information was added to the manuscript. 

"This work was supported by T3: Texas A&M Triads for Transformation [grant numbers 1762, 2020], Texas A&M University Institutional Funding."

This information was added in the disclosure statement. 

"This work was supported by T3: Texas A&M Triads for Transformation [grant numbers 1762, 2020], Texas A&M University Institutional Funding."

This information was added to our cover letter. 

We uploaded our dataset to the Texas Data Repository and shared with the public. The data is available at https://doi.org/10.18738/T8/YQ1XDX. 

7. We note that you have stated that you will provide repository information for your data at acceptance. Should your manuscript be accepted for publication, we will hold it until you provide the relevant accession numbers or DOIs necessary to access your data. If you wish to make changes to your Data Availability statement, please describe these changes in your cover letter and we will update your Data Availability statement to reflect the information you provide.

The reference for the dataset: 

Zhou, Xuan, 2023, "RDM Needs assessment", https://doi.org/10.18738/T8/YQ1XDX, Texas Data Repository, V1

We added the caption for the supporting information at the end of our manuscript.

All references cited in the text are in the reference list. None of the articles have been retracted to our knowledge. 

Reviewer 1

It is a well-structured work that helps to understand some of the challenges faced by new researchers who, sometimes, put their investigations at risk due to a lack of these skills. Thank you. 

About data - The data should be provided as part of the manuscript or its supporting information, or deposited to a public repository. If there are restrictions on publicly sharing data, those must be specified. 

Information about the underlying data was added. 

Reviewer 2 

The manuscript addresses a significant problem in the selected area of research. Furthermore, the contents are managed coherently. However, the following observations need redressal: Thank you. We addressed the comments one by one. 

1. Instead of mentioning 'Research Question 1" in the subheading in Results and Discussion, thematic headings would make them more comprehensible. This applies to all the subheadings in this section. 

We deleted “research question” from the subheads and added thematic headings for each level.

2. Discussion on the interview data is limited and needs further instances, elaboration, and due support from the reviewed literature. 

We elaborated on the discussion of the interview data and better connected the discussion with the literature review. 

3. It seems that the data has not properly been triangulated. 

This is a mixed-method study. We used interview data to supplement the information we obtained from the survey, but did not triangulate the interview data separately.

We added additional discussion about the interview data. 

4. Recommendations for future studies are not specified. 

Future research directions were added to the first paragraph of the Conclusion section. 

5. The authors might be advised to get the manuscript proofread by an expert in the English language. 

A native English speaker proofread and edited the manuscript.

---

## [Editor Report · Decision Letter 1]

8 Feb 2023

Research data management needs assessment for social sciences graduate students: A mixed methods study

PONE-D-22-30504R1

Dear Dr.,

We’re pleased to inform you that your manuscript has been judged scientifically suitable for publication and will be formally accepted for publication once it meets all outstanding technical requirements.

Kind regards,

Omar M Khraisat, Associate Professor

Academic Editor

PLOS ONE
---

## [Editor Report · Acceptance letter]

13 Feb 2023

PONE-D-22-30504R1 

Research data management needs assessment for social sciences graduate students: A mixed methods study 

Dear Dr. Xu:

I'm pleased to inform you that your manuscript has been deemed suitable for publication in PLOS ONE. Congratulations! Your manuscript is now with our production department. 

Kind regards, 

on behalf of

Dr. Omar M Khraisat 

Academic Editor

PLOS ONE